## [Peer Review File · Nature Communications]

Reviewers' Comments:

Reviewer #1:

Remarks to the Author:

The authors report the synthesis of chiral indole derivatives via a enantio- and diastereoselective method, using an NHC-iron-based C-H activation strategy. The racemic reaction using similar conditions was previously reported by Yoshikai (*Org Lett* 2015, 17, 442 – reference 55 in the current manuscript) and a first enantioselective version of this reaction was also previously explored by L. Ackermann (*Angew Chem Int Ed* 2017, 14197 – reference 15 in the current manuscript). In the current new work, the authors have further screened a wide family of chiral NHCs and found that a subtle change at the ligand system i.e. replacing a Me substituent in the ligand used for their previous study (L1 here) with an OMe (L5) determined important changes in reaction selectivity. In their mechanistic discussion, the authors rationalise that a Fe-OMe interaction plays an important role in the enantioselectivity determining (turnover limiting) transition state. The authors explore a wide range of substrates, all of which convert with excellent selectivities. The analyses of the organic products is done in an excellent manner (including chromatograms and a detailed description of the experimental procedure). A proposed mechanism was further devised by exploring the potential energy surface (PES) of the C-H activation reaction. Here, the authors have determined that the reaction takes place initially on a triplet surface, but crosses "back-and-forth" to a quintet surface. The entry in the catalytic cycle is an (NHC)Fe(sytene)₂ complex. A model of this complex using the simpler (achiral) IMes NHC was obtained and characterised by X-ray diffraction and by Mössbauer spectroscopy. All in all, this study was done at high technical standard, and the method is clearly an improvement of what is known in this literature for this specific reaction. Examples of enantioselective transformations with iron are rare, and this work definitely enhances the knowledge in the field. Nevertheless, because of strong similarities with previous published work (references 15 and 55), I have some reservations with respect to suitability in *Nature Communications*. A number of comments and suggestions are given below.

1) The scholar presentation of this work can be improved with respect to referencing the relevant previous work. While the relevant references are given in the paper, the authors should make clear in the introduction and in Figure 1 what is already known about this reaction – and here I mention again the racemic version (reference 55) and the authors' own previous studies on this exact reaction using almost identical reaction conditions (reference 15).

2) While I get the reference of "escaping flatland" and designing 3D molecules, I am afraid it is not really clear from the introduction what this means and why it is important. Maybe the authors should focus on this aspect, rather than on how benign and non-toxic iron is (a feature which I guess is now clear to the synthetic community). Moreover, while the strategy used here is interesting, atroposelective iron couplings (albeit C–C couplings) are known (see latest example Smith, *Nat Chem* 2023, 15, 357).

3) Computational chemistry part: The calculations are performed on a fairly good theory level, which is very appreciated, especially given the size of the system investigated. Nevertheless, I would caution the authors not to overinterpret differences less than 5 kcal/mol especially when dealing with open-shell iron species. This is within the limit of DFT and should be taken into account. I would suggest that it is impossible to differentiate between TS9 – Int6 pathway and TS3-Int4-TS5-Int6 pathway. In this respect, I would be also very cautious in interpreting secondary interactions especially when looking at open-shell transition states when using the non-range separated DFT (B3LYP). I would strongly recommend either, investigating this aspect with more caution (i.e. CASSCF, energy decomposition analysis, etc) or taking the reference to weak dispersion interactions out of the title.

4) Still referring to the computational chemistry part: looking at the reaction path the authors propose, I cannot help not noticing that the Int1 at the end of the PES (Fig 6) is 6.1 kcal/mol lower in energy than the same Int1 at the beginning of the reaction. Could it be that what the authors meant to represent is the Fe - sigma-bonded C-C product after RE, before dissociation? Looking at the same representation in the SI, the Int6 (in blue, 3kcal/mol) should be denoted as triplet, not quintet.

5) This work reports a very elegant freeze-quench study, consistent with the authors' (M. Neidig) excellent work in the field. Study allows the observation of the direct activation product of Fe(acac)₃ CyMgBr, followed by the reaction of the resulting species with (NHC)H⁺, to give the (NHC)Fe(styrene)₂ species. This was nicely demonstrated for the achiral IMes model, and was

further supported by crystallographic studies. I wonder, nevertheless, if the authors have attempted to use the crystal structure of B* and to calculate the Mössbauer parameters, in order to see if this matches the spin state they use for their DFT modelling. This should be done fairly easily and it would add more value to the computational studies.

6) The same computational chemistry approach can be used to decipher that rather complex mixture that forms if the analogous reaction is performed between A and L5. The Mössbauer spectrum of the reaction products allows to fit a species which has the same parameters as B*, nevertheless, this seems to be less than 50% of what is being formed. Did the authors have attempted to fit other species, or have a rough guess on what these species may be? Since the authors (Ackermann et al) discussed the importance of meta-substituted aryls in NHCs in a previous publication, I was wondering if a 2,4-substituted Aryl-based NHC could also be used in the model study. It could be that the lack of selectivity arises from inefficient shielding of these types of carbenes.

7) While mixing A and L5 gives a complex reaction mixture, conducting the same reaction in the presence of the substrate, gives a fairly selective reaction (75% - based on the Mössbauer spectrum, figure 5b.) Nevertheless, the authors do not comment on what this species may be. I find this aspect really interesting, as this species would be the on or off cycle resting state of the catalytic reaction. Here, the authors could again use a combination of the computational studies and Mössbauer spectroscopy to assign these species. I notice that the aliquot was taken 6 hours into the catalytic reaction. Does the composition of the iron-containing species stay the same over the course of the reaction?

8) Perhaps I missed it entirely, but I could not find the experimental procedures and characterisation data for the metal-based complexes (A, B* and B) in the supporting information. Since A and B* could be isolated, an NMR spectrum and a precise experimental procedure should be given. Moreover, the magnetic measurements (Evans' or SQUID) would indicate if the ground state is indeed singlet, triplet or quintet.

To summarise, I find this work of excellent technical quality. Nevertheless, number of mechanistic questions still remain open, and can be addressed using a combination of freeze-quenched Mössbauer spectroscopy and computational chemistry. With regards to the suitability of the study for Nature Communications (especially in the light of the catalytic application chosen), I am not convinced that the advance made in this manuscript compared to the ACIE paper from 2017 (ref 15) is significant to warrant acceptance.

Reviewer #2:

Remarks to the Author:

In this manuscript, the authors described an interesting approach to double stereoselective C–H activation using an iron catalyst. This method allows for the generation of substituted indoles with two chiral elements. Notably, the iron-catalyzed enantio-selective C–H alkylation can be conducted under very mild conditions at room temperature, resulting in highly uncommon enantioselective transformations. The key to achieving remarkable diastereo- and enantioselectivities lies in the unique design of a ligand featuring a remotely decorated N-heterocyclic carbene ligand. This ligand ensures the presence of secondary noncovalent dispersion interactions, enabling precise control over the C–N-axial chirality and C-stereogenic centers. Mechanistic studies, both experimental and computational, have revealed an iron (0) complex as the catalytically active species. It offers a valuable complementary approach for accessing axially chiral arylindoles. The authors have done an excellent job in presenting their manuscript, providing a clear context to illustrate the importance of their work, and adequately discussing their findings. This reviewer recommends publication of the manuscript in Nature Communications after addressing following issues.

1. Regarding the substrate scope illustrated in Figure 3, it appears that the styrene is crucial for the success of this transformation. What happens when other alkenes with alkyl or internal alkenes are introduced?

2. The enantioselectivity and yield experience a significant decrease when the ligand L2 is replaced with L4. What could be the primary reason for this drop?

3. In the Supplementary Information, it is necessary to re-collect the HPLC traces for compounds 29 and 34.

4. A typical review should be cited: Recent Advances in Catalytic Asymmetric Construction of Atropisomers. Chem. Rev. 2021, 121, 4805–4902.

Reviewer #3:

Remarks to the Author:

The authors report an iron-catalyzed directed alkylation of indoles with styrenes. By using a chiral NHC ligand and a 2-substituted aryl N-substituent on the substrate, the authors could control both the stereoselectivity of the incoming alkyl group, and the atroposelectivity in the resulting congested indole product. While the reaction scope is automatically restricted by the necessary use of an indole substrate possessing an imine directing group at the 3-position, an N-substituent (2-substituted arene), and a styrene coupling partner, a variety of 1,2,3-trisubstituted indoles were obtained with moderate to high yields, and high diastereo- and enantioselectivity. The authors also performed a mechanistic analysis by experiment and calculation, to reveal that the active species is probably a low-valent, 1:1 Fe-NHC complex that bears two alkene ligands, which activates the C-H bond through a LLHT mechanism. Interestingly, both the axial and point chirality are rationalized by favorable noncovalent interactions (CH- π and π - π) during the migratory insertion step. The SI is of high quality, methods are described in detail, and the compounds are well characterized.

One major criticism is the structure of the introduction, which results in overselling the novelty of this work. Thus, the introduction emphasizes the abundance, price, and non-toxicity of iron using references 56,57, which is rather common sense and does not require two charts in my opinion; but this is just my personal preference, what is more serious is the overly restrictive description of iron-catalyzed asymmetric transformations (Fig. 1b), lacking the authors' previous work (Ref. 15), which describes essentially the same reaction: Fe(acac)₃ and a slightly modified NHC ligand (MeO instead of Me), the same indole substrate (just the N-substituent is different), the same styrene coupling partners and same reaction conditions (CyMgX, TMEDA, THF, 45 C) to produce 1,2,3-trisubstituted indoles with high enantioselectivity. This key previous report is not described at all neither in the text, nor in Figure 1, and just buried among other references as reference 15: "This holds particular true for the most abundant, inexpensive and non-toxic (38) metal iron(13-16)." Thus, the main advance in the present work is just bulking up the N-substituent (benzyl -> 2-substituted aryl), which indeed results in an interesting dual chirality control. Moreover, while authors claim that "the approaches to such compounds being extremely rare and almost exclusively based on noble and toxic transition metals (41-48)", which is indeed generally true, they conveniently "forget" to mention or at least cite their own work on Co/NHC-catalyzed atroposelective directed arylation of similar indole substrates with chloroarenes (JACS 2022, 144, 798).

Other comments:

- the weak noncovalent interactions governing selectivity are one of the main selling points of the paper, even mentioned in the title. The authors should also quantify the extent of these interactions. And a minor issue: a higher resolution for the NCI plot is desirable
- General Procedure 4: the Schenck tube becomes vial during the procedure?
- dropwise addition of CyMgBr: does the addition rate influence the reactivity/selectivity?

Reply to comments by Reviewer 1

We appreciate **Reviewer 1** for favorable comments and many helpful suggestions!

Comments: The authors report the synthesis of chiral indole derivatives *via* an enantio- and diastereoselective method, using an NHC-iron-based C–H activation strategy. The racemic reaction using similar conditions was previously reported by Yoshikai (Org. Lett. 2015, 17, 442 – reference 55 in the current manuscript) and a first enantioselective version of this reaction was also previously explored by L. Ackermann (Angew. Chem. Int. Ed. 2017, 56, 14197 – reference 15 in the current manuscript). In the current new work, the authors have further screened a wide family of chiral NHCs and found that a subtle change at the ligand system i.e. replacing a Me substituent in the ligand used for their previous study (L1 here) with an OMe (L5) determined important changes in reaction selectivity. In their mechanistic discussion, the authors rationalize that a Fe-OMe interaction plays an important role in the enantioselectivity determining (turnover limiting) transition state. The authors explore a wide range of substrates, all of which convert with excellent selectivities. The analyses of the organic products is done in an excellent manner (including chromatograms and a detailed description of the experimental procedure). A proposed mechanism was further devised by exploring the potential energy surface (PES) of the C–H activation reaction. Here, the authors have determined that the reaction takes place initially on a triplet surface, but crosses “back-and-forth” to a quintet surface. The entry in the catalytic cycle is an (NHC)Fe(sytene)₂ complex. A model of this complex using the simpler (a-chiral) IMes NHC was obtained and characterized by X-ray diffraction and by Mössbauer spectroscopy.

All in all, this study was done at high technical standard, and the method is clearly an improvement of what is known in this literature for this specific reaction. Examples of enantioselective transformations with iron are rare, and this work definitely enhances the knowledge in the field. Nevertheless, because of strong similarities with previous published work (references 15 and 55), I have some reservations with respect to suitability in Nature Communications. A number of comments and suggestions are given below.

Question 1: The scholar presentation of this work can be improved with respect to referencing the relevant previous work. While the relevant references are given in the paper, the authors should make clear in the introduction and in Figure 1 what is already known about this reaction – and here I mention again the racemic version (reference 55) and the authors’ own previous studies on this exact reaction using almost identical

reaction conditions (reference 15).

Answer: We are grateful for the valuable suggestions from the reviewer. We have added a detailed description of the preliminary work (*Org. Lett.* **2015**, *17*, 442 and *Angew. Chem. Int. Ed.* **2017**, *56*, 14197) in the second paragraph of the introduction, and clearly represented the relevant content in Figure **1c**.

Question 2: While I get the reference of “escaping flatland” and designing 3D molecules, I am afraid it is not really clear from the introduction what this means and why it is important. Maybe the authors should focus on this aspect, rather than on how benign and non-toxic iron is (a feature which I guess is now clear to the synthetic community). Moreover, while the strategy used here is interesting, atroposelective iron couplings (albeit C–C couplings) are known (see latest example Smith, *Nat. Chem.* **2023**, *15*, 357).

Answer: We are grateful for the valuable suggestions from the reviewer. We have added content on the design and synthesis of 3D molecules with multiple chiral centers in the fourth paragraph of the introduction. In addition, we have deleted unnecessary descriptions of the advantages of iron and part of relevant content in Figure **1a**. Regarding the latest literature (*Nat. Chem.* **2023**, *15*, 357), it was published during our submission period and we have now added it to the references, see ref. 24.

Question 3: Computational chemistry part: The calculations are performed on a fairly good theory level, which is very appreciated, especially given the size of the system investigated. Nevertheless, I would caution the authors not to overinterpret differences less than 5 kcal/mol especially when dealing with open-shell iron species. This is within the limit of DFT and should be taken into account. I would suggest that it is impossible to differentiate between TS9–Int6 pathway and TS3–Int4–TS5–Int6 pathway. In this respect, I would be also very cautious in interpreting secondary interactions especially when looking at open-shell transition states when using the non-range separated DFT (B3LYP). I would strongly recommend either, investigating this aspect with more caution (i.e. CASSCF, energy decomposition analysis, etc) or taking the reference to weak dispersion interactions out of the title.

Answer: We thank the reviewer for the insightful comments. We agree that the oxidative addition pathway (TS3–int4–TS5–int6) cannot be completely excluded, therefore we have revised the manuscript and indicated the LLHT (TS9–int6) is most likely the optimal pathway for the generation of the alkyl iron(II) species **int6**, which

is more favorable than oxidative addition pathway by 4.7 kcal/mol.

To further confirm the importance of noncovalent interactions in controlling the stereoselectivity, we have performed energy decomposition analysis on the optimized geometries based on classical molecular force field (EDA-FF, *Mat. Sci. Eng. B*, **273**, 115425 (2021)) given the size of this system. For such purpose, the reductive elimination transition state was divided into three fragments (**Fig. R1a**), namely iron catalyst part (fragment 1), indole part (fragment 2) and alkyl part (fragment 3). The results of energy decomposition analysis show that the major component of noncovalent interaction in reductive elimination transition state **TS10** (*R_a*, *S*) is electrostatic interaction followed by dispersion interaction. The binding between fragment 1 and fragment 2 is the strongest among other combinations, which is also reflected in the atom-level electrostatic and dispersion interaction maps. The atoms that contribute most to the electrostatic interaction are iron from fragment 1 and C, N connected to iron in fragment 2 (**Fig. R1b**). The dispersion interaction also largely locates between fragment 1 and fragment 2 (**Fig. R1c**). The relative interaction energies of competing transition states in **Fig. R1d** demonstrate the dominant role of noncovalent interaction in controlling the stereoselectivity. Nevertheless, we have removed the weak dispersion interactions from the title.

Fig. R1 a) Energy decomposition analysis of the most favorable reductive elimination transition state **TS10 (R_a,S)** and interaction energy components between all fragments. b) Atom contribution map of electrostatic interaction in **TS10 (R_a,S)**. c) Atom contribution map of dispersion interaction in **TS10 (R_a,S)**. Trivial hydrogens are omitted for clarity. d) Relative free energies and interaction energies of reductive elimination transition states.

Question 4: Still referring to the computational chemistry part: looking at the reaction path the authors propose, I cannot help not noticing that the Int1 at the end of the PES (Fig 6) is 6.1 kcal/mol lower in energy than the same Int1 at the beginning of the reaction. Could it be that what the authors meant to represent is the Fe-sigma-bonded C-C product after RE, before dissociation? Looking at the same representation in the SI, the Int6 (in blue, 3kcal/mol) should be denoted as triplet, not quintet.

Answer: We thank the reviewer for this remind. The final elementary step of the PES in Figure 6 comprises the ligand exchange, where the product is released and iron(0)

complex **int1** is regenerated. The energy of -6.1 kcal/mol refers to the reaction system after ligand exchange, therefore we have changed the last species of the PES from **int1** into the product to avoid misunderstanding. Additionally, we have carefully checked the spin density for all the calculated species, and corrected the representation for **int6** in *Supplementary Information Figure S17*.

Question 5: This work reports a very elegant freeze-quench study, consistent with the authors' (M. Neidig) excellent work in the field. Study allows the observation of the direct activation product of Fe(acac)₃ and CyMgBr, followed by the reaction of the resulting species with (NHC)H⁺, to give the (NHC)Fe(styrene)₂ species. This was nicely demonstrated for the achiral IMes model, and was further supported by crystallographic studies. I wonder, nevertheless, if the authors have attempted to use the crystal structure of **B*** and to calculate the Mössbauer parameters, in order to see if this matches the spin state, they use for their DFT modelling. This should be done fairly easily and it would add more value to the computational studies.

Answer: We thank the reviewer for raising the question of the spin state of **B*** and their very useful suggestion of Mössbauer parameter calculations. **B*** is a $S = 1$ complex and the text has been revised according. This is based off of the Evans measurement of this complex which is consistent with $S = 1$ with a significant SOC contribution (now included in the *Supplementary Information Table S6*), prior detailed MCD studies of the ethylene analog of **B*** in ref. 63 which was also $S = 1$, and the suggested Mössbauer parameter calculations which are in agreement with this assignment (though with a degree of error quite common for such calculations). Noteworthy is that while **B** (assigned as the **L5** ligand analog of **B***) could not be isolated, Mössbauer calculations are also consistent with this complex having a $S = 1$ ground state and this has been included in the *Supplementary Information Table S6*, together with the Mössbauer calculations for **A**.

Question 6: The same computational chemistry approach can be used to decipher that rather complex mixture that forms if the analogous reaction is performed between **A** and **L5**. The Mössbauer spectrum of the reaction products allows to fit a species which has the same parameters as **B***, nevertheless, this seems to be less than 50% of what is being formed. Did the authors have attempted to fit other species, or have a rough guess on what these species may be? Since the authors (Ackermann et al) discussed the importance of meta-substituted aryls in NHCs in a previous publication, I was wondering if a 2,4-substituted Aryl-based NHC could also be used in the model study.

It could be that the lack of selectivity arises from inefficient shielding of these types of carbenes.

Answer: Unfortunately, the Mossbauer spectrum for the reaction of **A** and **L5** is quite complicated at both reaction time points, and we do not believe that an accurate multi-component fit is possible with the current data beyond the likely presence of a small amount of unreacted **A** that we can unambiguously fit. In the absence of structural information on the minor components which might not even have the NHC ligand bound, we do not believe we could confidently assign the minor species to any individual decomposition products from Mössbauer calculations. However, as noted by the reviewer in a subsequent comment the speciation is significantly cleaner during catalysis. This may indicate that a large excess of styrene is necessary to promote the formation of **B**, leading to more iron side product formation in the stoichiometric reaction. Text has been added to the *Supplementary Information* Figure **S14** and **S15** caption regarding this point.

To verify the significance of *meta*-substituted aryls in NHCs, we carried out a model study using 2,4-substituted NHC with adamantyl group at the *para*-position of the aromatic ring. The enantio-determining reductive elimination step is unsurmountable under the room temperature with a barrier of 34.3 kcal/mol (**Fig. R2a**). Additionally, we have performed control experiments applying 2,5-substituted NHC *versus* 2,4-substituted NHC (see **Fig. R2b**). When *meta*-substituted aryl-based NHC **L8** was used as chiral ligand, the target product could be obtained in excellent yield and stereoselectivity. However, when 2,4-substituted aryl-based NHC **L10** was employed, the yield and enantioselectivity of the reaction dropped significantly. Since the yield of this reaction was only 8%, the diastereoselectivity was difficult to determine.

a) Reductive elimination using 2,4-substituted NHC ligand

b) Control experiments

Fig. R2 a) Reductive elimination process when 2,4-substituted NHC ligand was used. b) Control experiments.

(4*R*,5*R*)-1,3-Bis(4-(*tert*-butyl)-2-methoxyphenyl)-4,5-diphenyl-4,5-dihydro-1*H*-imidazol-3-ium tetrafluoroborate (L10)

¹H NMR (400 MHz, CDCl₃) δ 8.87 (s, 1H), 7.48 – 7.34 (m, 10H), 7.17 (d, $J = 8.4$ Hz, 2H), 6.94 (d, $J = 1.9$ Hz, 2H), 6.89 (dd, $J = 8.4, 1.9$ Hz, 2H), 5.67 (s, 2H), 3.97 (s, 6H), 1.25 (s, 18H). **¹³C NMR (101 MHz, CDCl₃)** δ 158.39 (CH), 154.49 (C_q), 152.89 (C_q), 135.69 (C_q), 130.00 (CH), 129.74 (CH), 127.60 (CH), 125.90 (CH), 120.42 (C_q), 118.73 (CH), 109.43 (CH), 75.48 (CH), 56.15 (CH₃), 35.27 (C_q), 31.26 (CH₃). **¹⁹F NMR (377 MHz, CDCl₃)** δ -152.60, -152.66. **HRMS (ESI) m/z (M–BF₄)⁺**: calculated for (C₃₇H₄₃N₂O₂)⁺: 547.3319, found: 547.3321.

^1H NMR spectrum of L10

^{13}C NMR spectrum of L10

¹⁹F NMR spectrum of **L10**

Question 7: While mixing **A** and **L5** gives a complex reaction mixture, conducting the same reaction in the presence of the substrate, gives a fairly selective reaction (75% - based on the Mössbauer spectrum, figure 5b.) Nevertheless, the authors do not comment on what this species may be. I find this aspect really interesting, as this species would be the on or off cycle resting state of the catalytic reaction. Here, the authors could again use a combination of the computational studies and Mössbauer spectroscopy to assign these species. I notice that the aliquot was taken 6 hours into the catalytic reaction. Does the composition of the iron-containing species stay the same over the course of the reaction?

Answer: We apologise that the assignment of **B** was not clear. This is assigned to the chiral ligand analog of **B***, Fe(**L5**)(η^2 -styrene)₂ based on the similarity of the parameters of **B***, which is further supported by Mössbauer calculations (see response to **question 5** above). We have ensured this assignment is clear in the manuscript. We have taken additional reaction time points and the iron speciation remained unchanged.

Question 8: Perhaps I missed it entirely, but I could not find the experimental procedures and characterisation data for the metal-based complexes (**A**, **B*** and **B**) in the supporting information. Since **A** and **B*** could be isolated, an NMR spectrum and a precise experimental procedure should be given. Moreover, the magnetic measurements (Evans' or SQUID) would indicate if the ground state is indeed singlet, triplet or quintet.

Answer: We apologize for this oversight and have revised the *Supplementary Information* to include the requested procedures and characterization data for **A** and **B*** (see *Supplementary Information* Figure **S12** and **S13**). Note that **B** was not able to be isolated in this study but only observed spectroscopically. We have also included detailed procedures for the reaction studies performed in the *Supplementary Information*.

Comments: To summarise, I find this work of excellent technical quality. Nevertheless, number of mechanistic questions still remain open, and can be addressed using a combination of freeze-quenched Mössbauer spectroscopy and computational chemistry. With regards to the suitability of the study for Nature Communications (especially in the light of the catalytic application chosen), I am not convinced that the advance made in this manuscript compared to the ACIE paper from 2017 (ref 15) is significant to warrant acceptance.

Answer: Compared with our previous report, this work reports for the first time a possibility of controlling two chiral elements within one elementary step using the iron-catalyzed C–H activation. In the same lines, this is the first iron-based catalytic system, capable of introducing C–N atropisomerism (synthesis of C–N atropisomeric compounds is particularly challenging and implementation of the C–H activation methodology remains unprecedented). Although the substrates might look quite similar compared to the ones studied in 2017, from the fundamental viewpoint the catalyst needs to perfectly control an additional event, ie. stereinduction within the Ar–N unit, while perfectly orchestrating both stereoselective events, what translates into a very selective formation of one stereoisomer out of 4 possible stereoisomers. Such a reactivity is truly unique and therefore this work should not be considered as the extension of the substrate scope of the previous paper. The desired reactivity could be reached at room temperature by modifying the chiral NHC catalyst and optimizing the reaction conditions. During the design of new chiral NHC ligand (**L5**), we found that a Fe-OMe interaction plays an important role in the enantioselectivity determining transition state. The use of a catalytic amount of Grignard reagent proved that it only played a role in the generation of the catalytically active zero-valent iron complex and did not participate in the catalytic cycle. Regarding substrate expansion, electron-poor styrene derivatives with low enantioselectivity in previous reports can also be used in this work to obtain the desired products with excellent stereoselectivities. In addition, we also conducted in-depth reaction mechanism studies, where the zero-valent iron active intermediate was verified through Mössbauer spectroscopy. And computational

studies proposed a most likely LLHT reaction mechanism pathway. Therefore, we believe that this work is worthy of publication in *Nature Communications*.

Reply to comments by Reviewer 2

We appreciate **Reviewer 2** for favorable comments and many helpful suggestions!

Comments: In this manuscript, the authors described an interesting approach to double stereoselective C–H activation using an iron catalyst. This method allows for the generation of substituted indoles with two chiral elements. Notably, the iron-catalyzed enantioselective C–H alkylation can be conducted under very mild conditions at room temperature, resulting in highly uncommon enantioselective transformations. The key to achieving remarkable diastereo- and enantioselectivities lies in the unique design of a ligand featuring a remotely decorated N-heterocyclic carbene ligand. This ligand ensures the presence of secondary noncovalent dispersion interactions, enabling precise control over the C–N-axial chirality and C-stereogenic centers. Mechanistic studies, both experimental and computational, have revealed an iron (0) complex as the catalytically active species. It offers a valuable complementary approach for accessing axially chiral aryl-indoles. The authors have done an excellent job in presenting their manuscript, providing a clear context to illustrate the importance of their work, and adequately discussing their findings. This reviewer recommends publication of the manuscript in Nature Communications after addressing following issues.

Question 1: Regarding the substrate scope illustrated in Figure 3, it appears that the styrene is crucial for the success of this transformation. What happens when other alkenes with alkyl or internal alkenes are introduced?

Answer: We are grateful for the valuable suggestions from the reviewer. We have attempted various reactions using different alkyl or internal alkenes, but have not been able to obtain the desired products (see **Fig. R3**). We believe that the aromatic ring conjugated with the double bond plays a crucial role in the formation of the zero-valent iron complex $[\text{Fe}(\mathbf{L5})(\eta^2\text{-styrene})_2]$, which is essential for the progress of the reaction. Additionally, the significant steric hindrance of internal alkenes makes it difficult to coordinate to the metal center preventing the formation of the zero-valent iron complex.

Unsuitable olefins:

Fig. R3 Unsuitable olefins for the C–H alkylation.

Question 2: The enantioselectivity and yield experience a significant decrease when the ligand L2 is replaced with L4. What could be the primary reason for this drop?

Answer: We are grateful for the valuable suggestions from the reviewer. Comparing the reaction results of **L1**, **L2** and **L3**, we found that the introduction of a methoxy or a phenoxy group at the 4-position of the aryl group did not have a significant impact on the stereoselectivity of the reaction. However, removing the methoxyl group from 4-position to 3-position (**L4** vs. **L2**) narrows the chiral pocket causing steric congestion between methoxy group and indole substrate, which could be the main reason for the decrease of both enantioselectivity and yield (see **Fig. R4**).

Fig. R4 Enantiocontrol transition states using **L2** or **L4**.

Question 3: In the Supplementary Information, it is necessary to re-collect the HPLC traces for compounds **29** and **34**.

Answer: We have re-collected the HPLC traces for compounds **29** and **34** (see below).

(*R*, *S*)-1-(2-Isopropylphenyl)-2-(1-(*o*-tolyl)ethyl)-1*H*-indole-3-carbaldehyde (29)

The product was analyzed by HPLC to determine the enantiomeric excess: 99% e.e. (CHIRALPAK ID-3, *n*-hexane/*i*-PrOH = 95/5, flow rate: 1.0 mL/min, T = 20 °C, 250 nm), t_R (minor) = 10.98 min, t_R (major) = 14.90 min.

Peak #	RetTime [min]	Type	Width [min]	Area [mAU*s]	Height [mAU]	Area %
1	10.750	MM R	0.6830	381.91635	9.51910	49.8688
2	14.790	MM R	0.7690	383.92520	8.32140	50.1312

Peak #	RetTime [min]	Type	Width [min]	Area [mAU*s]	Height [mAU]	Area %
1	10.978	BB	0.2618	35.50762	1.61307	0.4645
2	14.898	BB	0.5043	7608.73682	222.86345	99.5355

(*R*, *R*)-2-(1-Ferrocenylethyl)-1-(2-isopropylphenyl)-1*H*-indole-3-carbaldehyde (34)

The product was analyzed by HPLC to determine the enantiomeric excess: 99% e.e. (CHIRALPAK ID-3, *n*-hexane/*i*-PrOH = 90/10, flow rate: 1.0 mL/min, T = 20 °C, 250 nm), t_R (minor) = 14.55 min, t_R (major) = 21.93 min.

Peak #	RetTime [min]	Type	Width [min]	Area [mAU*s]	Height [mAU]	Area %
1	14.413	BB	0.5413	2545.47754	70.99490	49.9916
2	21.911	BB	0.6028	2546.32886	57.48416	50.0084

Peak #	RetTime [min]	Type	Width [min]	Area [mAU*s]	Height [mAU]	Area %
1	14.545	BB	0.3635	65.69038	2.14371	0.5887
2	21.929	BB	0.6144	1.10923e4	266.06561	99.4113

Question 4: A typical review should be cited: Recent Advances in Catalytic Asymmetric Construction of Atropisomers. Chem. Rev. 2021, 121, 4805–4902.

Answer: We have added it to the references, see ref. 38.

Reply to comments by Reviewer 3

We appreciate **Reviewer 3** for favorable comments and many helpful suggestions!

Comments: The authors report an iron-catalyzed directed alkylation of indoles with styrenes. By using a chiral NHC ligand and a 2-substituted aryl N-substituent on the substrate, the authors could control both the stereoselectivity of the incoming alkyl group, and the atroposelectivity in the resulting congested indole product. While the reaction scope is automatically restricted by the necessary use of an indole substrate possessing an imine directing group at the 3-position, an N-substituent (2-substituted arene), and a styrene coupling partner, a variety of 1,2,3-trisubstituted indoles were obtained with moderate to high yields, and high diastereo- and enantioselectivity. The authors also performed a mechanistic analysis by experiment and calculation, to reveal that the active species is probably a low-valent, 1:1 Fe-NHC complex that bears two alkene ligands, which activates the C–H bond through a LLHT mechanism. Interestingly, both the axial and point chirality are rationalized by favorable noncovalent interactions (CH- π and π - π) during the migratory insertion step. The SI is of high quality, methods are described in detail, and the compounds are well characterized.

One major criticism is the structure of the introduction, which results in overselling the novelty of this work. Thus, the introduction emphasizes the abundance, price, and non-toxicity of iron using references 56,57, which is rather common sense and does not require two charts in my opinion; but this is just my personal preference, what is more serious is the overly restrictive description of iron-catalyzed asymmetric transformations (Fig. 1b), lacking the authors' previous work (Ref. 15), which describes essentially the same reaction: $\text{Fe}(\text{acac})_3$ and a slightly modified NHC ligand (MeO instead of Me), the same indole substrate (just the N-substituent is different), the same styrene coupling partners and same reaction conditions (CyMgX, TMEDA, THF, 45 °C) to produce 1,2,3-trisubstituted indoles with high enantioselectivity. This key previous report is not described at all neither in the text, nor in Figure 1, and just buried among other references as reference 15: "This holds particular true for the most abundant, inexpensive and non-toxic (38) metal iron(13-16)." Thus, the main advance in the present work is just bulking up the N-substituent (benzyl \rightarrow 2-substituted aryl), which indeed results in an interesting dual chirality control. Moreover, while authors claim that "the approaches to such compounds being extremely rare and almost exclusively based on noble and toxic transition metals (41-48)", which is indeed generally true, they conveniently "forget" to mention or at least cite their own work on

Co/NHC-catalyzed atroposelective directed arylation of similar indole substrates with chloroarenes (JACS 2022, 144, 798).

Answer: We are grateful for the valuable suggestions from the reviewer. We have added a detailed description of the preliminary work (*Org. Lett.* **2015**, *17*, 442 and *Angew. Chem. Int. Ed.* **2017**, *56*, 14197) in the second paragraph of the introduction, and clearly represented the relevant content in Figure **1c**.

We have added content on the design and synthesis of 3D molecules with multiple chiral centers in the fourth paragraph of the introduction. In addition, we have deleted unnecessary descriptions of the advantages of iron and part of relevant content in Figure **1a**.

We thank the reviewer for reminding us of the missing literature. Regarding the sentence "the approaches to such compounds being extremely rare and almost exclusively based on noble and toxic transition metals", it is used to describe the construction of the C–N axial chirality. And the work of Co/NHC still constructs C–C axis chirality, so we added it to the ref. 59. in the modified manuscript.

Compared with our previous report, this work reports for the first time a possibility of controlling two chiral elements within one elementary step using the iron-catalyzed C–H activation. In the same lines, this is the first iron-based catalytic system, capable of introducing C–N atropisomerism (synthesis of C–N atropisomeric compounds is particularly challenging and implementation of the C–H activation methodology remains unprecedented). Although the substrates might look quite similar compared to the ones studied in 2017, from the fundamental viewpoint the catalyst needs to perfectly control an additional event, ie. stereoinduction within the Ar–N unit, while perfectly orchestrating both stereoselective events, what translates into a very selective formation of one stereoisomer out of 4 possible stereoisomers. Such a reactivity is truly unique and therefore this work should not be considered as the extension of the substrate scope of the previous paper. The desired reactivity could be reached at room temperature by modifying the chiral NHC catalyst and optimizing the reaction conditions. During the design of new chiral NHC ligand (**L5**), we found that a Fe-OMe interaction plays an important role in the enantioselectivity determining transition state. The use of a catalytic amount of Grignard reagent proved that it only played a role in the generation of the catalytically active zero-valent iron complex and did not participate in the catalytic cycle. Regarding substrate expansion, electron-poor styrene derivatives with low enantioselectivity in previous reports can also be used in this work to obtain the desired products with excellent stereoselectivities. In addition,

we also conducted in-depth reaction mechanism studies, where the zero-valent iron active intermediate was verified through Mössbauer spectroscopy. And computational studies proposed a most likely LLHT reaction mechanism pathway. Therefore, we believe that this work is innovative and will provide guidance and assistance for the future developments in the field of iron-catalyzed asymmetric C–H activation.

Question 1: the weak noncovalent interactions governing selectivity are one of the main selling points of the paper, even mentioned in the title. The authors should also quantify the extent of these interactions. And a minor issue: a higher resolution for the NCI plot is desirable.

Answer: To quantify the noncovalent interactions, we have performed energy decomposition analysis on the optimized geometries based on classical molecular force field (EDA-FF, *Mat. Sci. Eng. B*, **273**, 115425 (2021)) given the size of this system. For such purpose, the reductive elimination transition state was divided into three fragments (**Fig. R1a**), namely iron catalyst part (fragment 1), indole part (fragment 2) and alkyl part (fragment 3). The results of energy decomposition analysis show that the major component of noncovalent interaction in reductive elimination transition state **TS10 (R_a, S)** is electrostatic interaction followed by dispersion interaction. The binding between fragment 1 and fragment 2 is the strongest among other combinations, which is also reflected in the atom-level electrostatic and dispersion interaction maps. The atoms that contribute most to the electrostatic interaction are iron from fragment 1 and C, N connected to iron in fragment 2 (**Fig. R1b**). The dispersion interaction also largely locates between fragment 1 and fragment 2 (**Fig. R1c**). The relative interaction energies of competing transition states in **Fig. R1d** demonstrate the dominant role of noncovalent interaction in controlling the stereoselectivity. Nevertheless, we have removed the weak dispersion interactions from the title.

Additionally, we have improved the resolution of the NCI plot in Figure 6.

a) Energy decomposition analysis

	Electrostatic (kcal/mol)	Repulsion (kcal/mol)	Dispersion (kcal/mol)
Frag1 and Frag2	-177.1	52.5	-43.3
Frag1 and Frag3	-128.9	6.2	-9.1
Frag2 and Frag3	84.7	111.2	-32.0

b) Atom contribution to electrostatic interaction

c) Atom contribution to dispersion interaction

d) Relative free energies and interaction energies of reductive elimination transition states

	$\Delta\Delta G^\ddagger$ (kcal/mol)	$\Delta\Delta E_{\text{int}}^\ddagger$ (kcal/mol)
TS10 (R_a,S) major product	0.0	0.0
TS10 (R_a,R) minor product	3.5	1.7
TS10 (S_a,S) minor product	5.1	2.9
TS10 (S_a,R) minor product	4.5	4.4

Fig. R1 a) Energy decomposition analysis of the most favorable reductive elimination transition state **TS10 (R_a,S)** and interaction energy components between all fragments. b) Atom contribution map of electrostatic interaction in **TS10 (R_a,S)**. c) Atom contribution map of dispersion interaction in **TS10 (R_a,S)**. Trivial hydrogens are omitted for clarity. d) Relative free energies and interaction energies of reductive elimination transition states.

Question 2: General Procedure 4: the Schenck tube becomes vial during the procedure?

Answer: We have changed the “vial” in the manuscript and *Supplementary Information* to “Schenck tube”.

Question 3: Dropwise addition of CyMgBr: does the addition rate influence the reactivity/selectivity?

Answer: We conducted controlled experiments on the rate of adding the Grignard reagent. Dropwise addition of Grignard reagent within 1-5 minutes has no significant

impact on the reaction results. Rapid addition of Grignard reagent (within a few seconds) led to a decrease in reaction yield and a slight reduction in stereoselectivity. Slow addition of the Grignard reagent (more than 10 minutes) resulted in the precipitation of magnesium salts in the syringe, leading to a decrease in reaction yield.

Reviewers' Comments:

Reviewer #1:

Remarks to the Author:

The authors submit a revised version of their manuscript where they described the installation of two chiral centres using iron catalysis. The transformation is catalysed by chiral NHC complexes and is applied to a variety of indole substrates. The manuscript is accompanied with a detailed reply to the reviewers' remarks and improvements of the technical quality of their manuscript. The authors have address in part my concerns. I appreciate especially the effort the authors have made in improving the Mössbauer part of their paper and the computational modelling of the Mössbauer parameters. Nevertheless, a few points still remain to be resolved.

(1) As also mentioned by the 3rd reviewer, the authors did not reference clearly their own previous work in the field in the original submission. There, they used a closely related catalyst and a closely related substrate class. In this revised version, this is made more clear, but I am still a bit disappointed on how the authors depicted this in Figure 1c. Namely, they left the catalyst structure out and they chose a ferrocenyl-based substrate which is most dissimilar to the scope in the current paper, even though the vast majority of substrates in their previous submission (ACIE) resembles the ones explored here. I think it should be clear to the reader what the advances here are.

(2) Regarding the computational chemistry part: I have mentioned in my original report that the authors should exercise caution when interpreting energy differences of less than 5 kcal/mol for open-shell systems, especially when discussion transition states. Therefore, I still could not derive from the manuscript why preference is not given for the oxidative addition pathway instead of the LLHT pathway.

(3) The authors have used EDA-FF to extract more information about the secondary interactions. I am not 100% this is the right approach, especially since Force Field approaches (as far as I know) do not take the spin component into account.

(4) Regarding the KIE measurements the authors have performed: looking at the plot in the SI (Figure S1), I notice that the authors have a linear dependence between time and yield, indicating a zeroth order reaction for the reaction. This does not come very intuitive to me. I also did not find anywhere the number of repetitions for these kinetic experiments, which is a must, especially since one infers here a secondary KIE (1.2). Probably, this secondary KIE allows the authors to differentiate between a C-H activation scenario and a LLHT scenario but I see no mention of this in the manuscript (in the computational chemistry part or mechanistic studies).

(5) Typographical errors: competition experiment instead of competitive (in the paragraph above Fig. 4). "The coordination in combination", not "in combine" – In the computational chemistry part. For the IR stretching frequencies in the SI, the gamma letter should be replaced by the Greek letter nu. ¹³C chemical shifts should be given with only one significant digit.

(6) For the Mössbauer part, it would also be nice to have the residual after the fit, so the reader can judge how good the fit was.

Reviewer #2:

Remarks to the Author:

The authors have addressed most of points raised. I would like to recommend the publication of this manuscript in Nature Communications without any delay.

Reviewer #4:

Remarks to the Author:

I was asked to specifically comment on the computational aspects of this work. The authors use a dispersion corrected density functional with a triple-zeta basis set. This is good. I am not sure about the B3LYP functional, though, especially when it comes to different spin states.

At the moment, I would say that the overall mechanism is sufficiently supported by the data. However, when it comes to the specific claims made about (i) the spin-state, (ii) the reaction path taken (TS3 vs TS9), and (iii) the reasons for enantioselectivity, I do not think that these are

sufficiently supported by the data. Getting accurate results on all of these is very hard and requires significant additional computations (see below). The authors should either make it clear that these are tentative results or perform appropriate additional computations.

1. Regarding the spin-state, is there a good reason why the Fe(II) compounds should not be low-spin singlets? When the authors report singlet, triplet, quintet energies in Figure S17, are these all computed for the same geometry? To make a fair comparison, the authors have to re-optimize the structures for the given spin state and then run the B3LYP/def2-TZVPP single points.

2. It would be good to know how robust the results are with respect to different functionals. The authors should add computations using two or three entirely different functionals (e.g. wB97X-V, M11-L, and TPSSH) to see how much the results change. Alternatively, they could run DLPNO-CCSD(T) calculations. If the spin states and barrier heights are largely unaltered between these different methods, then I would be inclined to trust the results.

3. It would also be important to see the individual influences on the relative energies (double-zeta vs triple-zeta, solvation, zero-point energies, entropy). Are the relative barrier heights always similar?

4. The authors should contrast their results to another recent paper on the stereoselective alkylation of indoles [Adv. Synth. Catal. 2023, DOI: 10.1002/adsc.202300845]. This recent paper found that differential solvation effects and not steric interactions were the main contribution to stereoselectivity.

5. I do think the EDA-FF analysis is fine and I agree that the results should be independent from the spin on the iron atom. But I would only trust EDA-FF if the overall energetics of the four TS10 versions are consistent when computed with different functionals.

6. I don't think the presentation of Figure 6 is ideal. The numbers and labels are very small. It should be possible to increase the font size.

Reply to comments by Reviewer 1

We appreciate **Reviewer 1** for favorable comments and many helpful suggestions!

Comments: The authors submit a revised version of their manuscript where they described the installation of two chiral centers using iron catalysis. The transformation is catalyzed by chiral NHC complexes and is applied to a variety of indole substrates. The manuscript is accompanied with a detailed reply to the reviewers' remarks and improvements of the technical quality of their manuscript. The authors have address in part my concerns. I appreciate especially the effort the authors have made in improving the Mössbauer part of their paper and the computational modelling of the Mössbauer parameters. Nevertheless, a few points still remain to be resolved.

Question 1: As also mentioned by the 3rd reviewer, the authors did not reference clearly their own previous work in the field in the original submission. There, they used a closely related catalyst and a closely related substrate class. In this revised version, this is made more clear, but I am still a bit disappointed on how the authors depicted this in Figure 1c. Namely, they left the catalyst structure out and they chose a ferrocenyl-based substrate which is most dissimilar to the scope in the current paper, even though the vast majority of substrates in their previous submission (ACIE) resembles the ones explored here. I think it should be clear to the reader what the advances here are.

Answer: We are grateful for the valuable suggestions from the reviewer. We have added the description of the alkene scope and chiral ligand structure of preliminary work in **Figure 1c**.

Question 2: Regarding the computational chemistry part: I have mentioned in my original report that the authors should exercise caution when interpreting energy differences of less than 5 kcal/mol for open-shell systems, especially when discussion transition states. Therefore, I still could not derive from the manuscript why preference is not given for the oxidative addition pathway instead of the LLHT pathway.

Answer: We have used various functionals to test the robustness of the computational results that give preference to ligand-to-ligand hydrogen transfer (LLHT) pathway. **Figure R1** shows the energy profiles of the competing oxidative addition and LLHT pathways under different functionals. Functionals including TPSSh, ω -B97XD and PBE0 all indicate LLHT is more favorable than oxidative addition by 5.9 kcal/mol to 7.7 kcal/mol. Furthermore, the most stable spin state of every species largely remains unchanged and are high spin state. As other functionals lead to exclusive consistent

results, we believe LLHT is most likely the preferred pathway for the generation of the alkyl iron(II) intermediate. And the results of this test have been included in the *Supplementary Information Figure S20*.

Figure R1. Free energy profiles of the competing oxidative addition and LLHT pathways using various functionals.

Question 3: The authors have used EDA-FF to extract more information about the secondary interactions. I am not 100% this is the right approach, especially since Force Field approaches (as far as I know) do not take the spin component into account.

Answer: Considering the number of atoms in this system, we chose EDA-FF which could tolerate the molecular size and give the results in a cheap and fast manner to perform detail investigation of secondary interactions. EDA-FF is performed based on the geometry and atomic charges which are calculated by Gaussian software considering the multiplicity. Thus, along with reviewer 4, we believe EDA-FF analysis is suitable for our system.

Question 4: Regarding the KIE measurements the authors have performed: looking at the plot in the SI (Figure S1), I notice that the authors have a linear dependence between time and yield, indicating a zeroth order reaction for the reaction. This does not come very intuitive to me. I also did not find anywhere the number of repetitions for these kinetic experiments, which is a must, especially since one infers here a secondary KIE (1.2). Probably, this secondary KIE allows the authors to differentiate between a C–H activation scenario and a LLHT scenario but I see no mention of this in the manuscript (in the computational chemistry part or mechanistic studies).

Answer: We are grateful for the valuable suggestions from the reviewer. We repeated the KIE experiment and found that k_H/k_D was close to 1.2 in both measurements (see **Figure R2a and R2b**). Since we only selected a few data points at the beginning of the reaction when conducting the KIE experiment, and the reaction rate will decrease over time (see *Supplementary Information Figure S5 and S9*), we do not think that there is a linear dependence between time and yield. According to the result of KIE experiment ($k_H/k_D < 2$), we believe that the C–H cleavage is facile and is unlikely to be the rate-determining step of the reaction. Regarding whether the reaction goes through the oxidative addition pathway or the LLHT pathway, we infer that the LLHT is most likely the preferred pathway based on the results of DFT calculations.

Kinetic isotope effect experiment

The kinetic isotope effect (KIE) was examined by applying the initial rate method.

To a flame-dried and N₂-purged Schlenk tube were added indole substrate **1a** (0.2 mmol, 73.6 mg), Fe(acac)₃ (0.02 mmol, 7.0 mg) and chiral NHC ligand **L5** (0.02 mmol, 15.8 mg). The Schlenk tube was then sealed, purged and backfilled with N₂ three times. Ethyl ether (0.5 mL), TMEDA (0.4 mmol, 60 μL), 4-fluorostyrene **2b** (0.3 mmol, 36 μL) and fluorobenzene (0.2 mmol, 19 μL) were added *via* syringe. CyMgBr (1 M in THF, 0.1 mmol, 0.1 mL) was then added dropwise and the resulting mixture was stirred at room temperature (*t* = 0 min). Aliquots (50 μL) were removed periodically every 20 min. The conversion was determined by ¹⁹F NMR using fluorobenzene as the internal standard.

The same procedure was applied with indole substrate [D]₁-**1a** (Note: the last point was not taken into account due to reduction of the reaction rate).

The following results were obtained.

Table R1a. Kinetic isotope effect

Entry	Time (min)	Yield (%) (with 1a)	Yield (%) (with [D] ₁ - 1a)
1	20	1	0
2	40	3	2
3	60	5	3.5
4	80	6.5	5
5	100	9	7
6	120	11	7.5

Figure R2a. Kinetic isotope effect

Table R1b. Kinetic isotope effect

Entry	Time (min)	Yield (%) (with 1a)	Yield (%) (with [D] ₁ - 1a)
1	20	0.5	0
2	40	2.5	2.5
3	60	4	4
4	80	6.5	5
5	100	8	7
6	120	9.5	8

**Figure R2b.** Kinetic isotope effect

Question 5: Typographical errors: competition experiment instead of competitive (in the paragraph above Fig. 4). “The coordination in combination”, not “in combine” – In the computational chemistry part. For the IR stretching frequencies in the SI, the gamma letter should be replaced by the Greek letter nu. ¹³C chemical shifts should be given with only one significant digit.

Answer: We thank the reviewer for the careful review. We adapted the revised manuscript and *Supplementary Information* accordingly.

Question 6: For the Mössbauer part, it would also be nice to have the residual after the fit, so the reader can judge how good the fit was.

Answer: We are grateful for the valuable suggestions from the reviewer. We have added the corresponding residuals to the Mössbauer spectra in the *Supplementary Information* (see *Supplementary Information* **Figure S12 – S16**).

Reply to comments by Reviewer 4

We appreciate **Reviewer 4** for favorable comments and many helpful suggestions!

Comments: I was asked to specifically comment on the computational aspects of this work. The authors use a dispersion corrected density functional with a triple-zeta basis set. This is good. I am not sure about the B3LYP functional, though, especially when it comes to different spin states.

At the moment, I would say that the overall mechanism is sufficiently supported by the data. However, when it comes to the specific claims made about (i) the spin-state, (ii) the reaction path taken (TS3 vs TS9), and (iii) the reasons for enantioselectivity, I do not think that these are sufficiently supported by the data. Getting accurate results on all of these is very hard and requires significant additional computations (see below). The authors should either make it clear that these are tentative results or perform appropriate additional computations.

Question 1: Regarding the spin-state, is there a good reason why the Fe(II) compounds should not be low-spin singlets? When the authors report singlet, triplet, quintet energies in Figure S17, are these all computed for the same geometry? To make a fair comparison, the authors have to re-optimize the structures for the given spin state and then run the B3LYP/def2-TZVPP single points.

Answer: The iron(II) complexes in the computed reaction profile contain electrophilic NHC ligand, which according to Crystal Field Theory, is a weak-field ligand causing a small splitting Δ_{oct} of the d -orbitals. Therefore, it is easier to put electrons into the higher energy set of orbitals than pairing two electrons into the same low-energy orbital that leading to repulsion energy. Furthermore, there are Fe–C(sp^2) and Fe–C(sp^3) σ -bondings in iron(II) complexes where iron center donates one electron, that could lead to higher spin states.

Figure R3. Crystal Field Diagram for iron(II) species taking int6 (quintet) as example.

Additionally, for each species in *Supplementary Information Figure S17*, triplet and quintet energies were acquired based on the structures after re-optimization with respective multiplicity. In Cartesian coordinates section, we included all the structures with different multiplicity.

Question 2: It would be good to know how robust the results are with respect to different functionals. The authors should add computations using two or three entirely different functionals (e.g. wB97X-V, M11-L, and TPSSh) to see how much the results change. Alternatively, they could run DLPNO-CCSD(T) calculations. If the spin states and barrier heights are largely unaltered between these different methods, then I would be inclined to trust the results.

Answer: We thank the reviewer for this valuable suggestion. We have used various functionals to test the robustness of the computational results presented in the manuscript. **Figure R1** shows the energy profiles of the competing oxidative addition and ligand-to-ligand hydrogen transfer (LLHT) pathways under different functionals. Functionals including TPSSh, ω -B97XD and PBE0 all indicate LLHT is more favorable than oxidative addition by 5.9 kcal/mol to 7.7 kcal/mol, which reach the same results with B3LYP. Furthermore, the most stable spin state of every species largely remains unchanged and are high spin state. The results of this computational methods test have been included in the *Supplementary Information Figure S20*.

Figure R1. Competing oxidative addition and LLHT processes using various functionals.

Question 3: It would also be important to see the individual influences on the relative energies (double-zeta vs triple-zeta, solvation, zero-point energies, entropy). Are the relative barrier heights always similar?

Answer: We have used triple zeta and double zeta basis sets to calculate relative energies between LLHT transition state TS9 and oxidative addition transition state TS3. The two basis sets give similar relative energies and favor LLHT pathway.

Figure R4. Relative energies of LLHT transition state TS9 and oxidative addition transition state TS3 using triple zeta and double zeta basis sets.

Question 4: The authors should contrast their results to another recent paper on the stereoselective alkylation of indoles [Adv. Synth. Catal. 2023, DOI: 10.1002/adsc.202300845]. This recent paper found that differential solvation effects and not steric interactions were the main contribution to stereoselectivity.

Answer: We didn't observe obvious solvent effect during the optimization of reaction conditions. Changing the solvent from THF to 2-MeTHF, toluene or Et₂O resulted in similar diastereomeric ratio and enantiomeric excess (see *Supplementary Information Table S2* for details). Therefore, we believe solvent effect has minor contribution to the stereoselectivity.

Table R2. Optimization of reaction conditions.

Entry	Solvent	Yield (%)	d.r.	e.e. (%)
1	THF	64	>95:5	95
2	2-MeTHF	63	>95:5	94
3	Toluene	68	>95:5	95
4	Et ₂ O	74	>95:5	96

Reaction conditions: **1a** (0.1 mmol), **2a** (0.15 mmol), Fe(acac)₃ (10 mol%), **L5** (20 mol%), CyMgX (1 M in THF, 0.11 mmol) and TMEDA (0.2 mmol) were stirred in solvent (0.2 mL) at T °C for 72 h under N₂, then added HCl aq. (1 M, 1.0 mL) and stirred for 2 h. The yield was determined by ¹H NMR spectroscopy using 1,3,5-trimethoxybenzene as the internal standard (yield of isolated product given within parentheses). The diastereomeric ratio (d.r.) was determined by ¹H NMR spectroscopy. The enantiomeric excess (e.e.) was determined by HPLC.

Question 5: I do think the EDA-FF analysis is fine and I agree that the results should be independent from the spin on the iron atom. But I would only trust EDA-FF if the overall energetics of the four TS10 versions are consistent when computed with different functionals.

Answer: We thank the reviewer for this comment. We have used different functionals to compute stereoselectivity and the relative free energies between the four competing reductive elimination transition states are largely consistent.

Figure R5. DFT-computed stereoselectivity using different functionals.

Question 6: I don't think the presentation of Figure 6 is ideal. The numbers and labels are very small. It should be possible to increase the font size.

Answer: Thank the reviewer for this helpful comment. We have increased the font size of numbers and labels in **Figure 6**.

Reviewers' Comments:

Reviewer #1:

Remarks to the Author:

In the current revised version, the authors address most of my concerns and the presentation of the material is more fair for the reader, especially in the context of previous work that was performed (Figure 1c vs 1e).

While there are some mechanistic subtleties which can still be discussed, I believe the manuscript is publishable as it is and the readers can decide for themselves of specific aspects. I was asked initially to comment on the Mössbauer part, which was done at a very high level from the beginning.

I am still not convinced about the arguments brought by the authors to reviewer 4, which contain some factual errors. For example, the answer to question 1: (1) the complex is octahedral, not tetrahedral, hence the splitting diagram is wrong, (2) Fe(II) is d6, not d5 (3) NHCs are strong field ligands due to their pi-accepting properties, (4) Crystal Field Theory cannot be used in any way for ligands with pi-contributions (5) there are examples of low-spin Fe species which are tetrahedral, the most famous example being Fe(nornornyl)₄, (6) if a closed shell spin state is given in the input, the authors may very well find a geometry in between tetrahedral and square planar.

Reviewer #4:

Remarks to the Author:

The authors have performed extended additional computations. I think it is now fair to say that the results are robust with the level of theory, and that the main conclusions also hold with different functionals. Fig. 6 is now sufficiently supported by the data. I am unable to comment on other aspects of the work.

There is one point of confusion left. In Figure R3, the authors present the electron configuration related to a sextet in an Fe(III) complex. This has nothing to do with the Fe(II) quintets discussed in the paper. I trust that this was just random oversight.

Reply to comments by Reviewer 1

We appreciate **Reviewer 1** for favorable comments and many helpful suggestions!

Comments: In the current revised version, the authors address most of my concerns and the presentation of the material is fairer for the reader, especially in the context of previous work that was performed (Figure 1c vs 1e).

While there are some mechanistic subtleties which can still be discussed, I believe the manuscript is publishable as it is and the readers can decide for themselves of specific aspects. I was asked initially to comment on the Mössbauer part, which was done at a very high level from the beginning.

Question 1: I am still not convinced about the arguments brought by the authors to reviewer 4, which contain some factual errors. For example, the answer to question 1: (1) the complex is octahedral, not tetrahedral, hence the splitting diagram is wrong, (2) Fe(II) is d6, not d5 (3) NHCs are strong field ligands due to their pi-accepting properties, (4) Crystal Field Theory cannot be used in any way for ligands with pi-contributions (5) there are examples of low-spin Fe species which are tetrahedral, the most famous example being Fe(nornornyl)4, (6) if a closed shell spin state is given in the input, the authors may very well find a geometry in between tetrahedral and square planar.

Answer: We are grateful for the valuable suggestions from the reviewer. Intermediate **int6** (quintet) is indeed tetrahedral as shown in the 3D geometry diagram (Figure R1). And we agree that using Crystal Field Theory is inappropriate in solving the spin state issue of iron(II) species. Upon investigating the geometries of different multiplicities of **int6**, we find that each spin state corresponds to a unique geometry. Specifically, the singlet **int6** adopts a quadrangular pyramidal configuration due to the coordination of the methoxy group, whereas the triplet **int6** exhibits a square planar geometry. We surmise that the tetrahedral geometry of quintet iron(II) species results in a favorable multiplicity of high spin states.

The discussion on the preference for high spin state iron(II) species was not the primary focus of the computational studies, therefore, it has not been included in either the manuscript or the Supplementary Information.

Figure R1. Relative energies for different multiplicity of intermediate **int6** and corresponding 3D geometries.

Reply to comments by Reviewer 4

We appreciate **Reviewer 4** for favorable comments and helpful suggestions!

Comments: The authors have performed extended additional computations. I think it is now fair to say that the results are robust with the level of theory, and that the main conclusions also hold with different functionals. Fig. 6 is now sufficiently supported by the data. I am unable to comment on other aspects of the work.

Question 1: There is one point of confusion left. In Figure R3, the authors present the electron configuration related to a sextet in an Fe(III) complex. This has nothing to do with the Fe(II) quintets discussed in the paper. I trust that this was just random oversight

Answer: We thank the reviewer for pointing out the oversight. For the favor of high spin state iron(II) species, we investigate geometries of different multiplicities of **int6**. We find that each spin state corresponds to a unique geometry, specifically, the singlet **int6** adopts a quadrangular pyramidal configuration due to the coordination of the methoxy group, whereas the triplet **int6** exhibits a square planar geometry. We surmise that the tetrahedral geometry of quintet iron(II) species results in a favorable multiplicity of high spin states.

The discussion on the preference for high spin state iron(II) species was not the primary focus of the computational studies, therefore, it has not been included in either the manuscript or the Supplementary Information.

Figure R1. Relative energies for different multiplicity of intermediate **int6** and corresponding 3D geometries.